# Anti-Migraine Effect of the Herbal Combination of Chuanxiong Rhizoma and Cyperi Rhizoma and UPLC-MS/MS Method for the Simultaneous Quantification of the Active Constituents in Rat Serum and Cerebral Cortex

**DOI:** 10.3390/molecules24122230

**Published:** 2019-06-14

**Authors:** Sha Wu, Li Guo, Feng Qiu, Muxin Gong

**Affiliations:** 1School of Traditional Chinese Medicine, Capital Medical University, Fengtai District, Beijing 100069, China; wusha729@163.com (S.W.); guoli183@126.com (L.G.); autumn3393@hotmail.com (F.Q.); 2Beijing Key Lab of TCM Collateral Disease Theory Research, Capital Medical University, Fengtai District, Beijing 100069, China

**Keywords:** migraine, Chuanxiong Rhizoma, Cyperi Rhizoma, active constituents, UPLC-MS/MS

## Abstract

Chuanxiong Rhizoma and Cyperi Rhizoma (CRCR), an ancient and classic formula comprised of Chuanxiong Rhizoma and Cyperi Rhizoma in a weight ratio of 1:2, has long been used for curing migraine. This study aimed to explore their anti-migraine effect and active constituents. A nitroglycerin (NTG)-induced migraine model in rats was established to evaluate pharmacological effects. Cerebral blood flow was detected by a laser Doppler perfusion monitor. The levels of endothelin-1 (ET-1), γ-aminobutyric acid (GABA), nitric oxide synthase (NOS), nitric oxide (NO), 5-hydroxytryptamine (5-HT), 5-hydoxyindoleacetic acid (5-HIAA), calcitonin gene-related peptide (CGRP) and β-endorphin (β-EP) were quantified with enzyme-linked immunosorbent assay. CGRP and c-Fos mRNA expression were quantified with quantitative real-time polymerase chain reaction. A UPLC-MS/MS method was developed and validated for the simultaneous quantification of active constituents in rat serum and cerebral cortex. CRCR significantly increased cerebral blood flow, decreased the levels of ET-1, GABA and NOS, and increased the levels of 5-HT, 5-HIAA and β-EP in NTG-induced migraine rats. CGRP levels and CGRP mRNA expression, as well as c-Fos mRNA expression in the brainstem were markedly down-regulated with the treatment of CRCR. After oral administration of CRCR, ferulic acid (FA), senkyunolide A (SA), 3-*n*-butylphthalide (NBP), *Z*-ligustilide (LIG), *Z*-3-butylidenephthalide (BDPH), cyperotundone (CYT), nookatone (NKT) and α-cyperone (CYP) were qualified in rat serum and cerebral cortex. The above results suggested that CRCR showed powerfully therapeutic effects on migraine via increasing the cerebral blood flow, decreasing the expression of CGRP and c-Fos mRNA, and regulating the releasing of ET-1, GABA, NOS, 5-HT, 5-HIAA, CGRP and β-EP in the serum and brainstem, consequently relieving neurogenic inflammation. The active constituents in CRCR for treating migraine were FA, SA, NBP, LIG, BDPH, CYT, NKT and CYP. These findings contributed for the further use of CRCR as a combinational and complementary phytomedicine for migraine treatment.

## 1. Introduction

Migraine is a chronic paroxysmal neurological disorder characterized by recurrent attacks of predominantly unilateral throbbing head pain [1]. The trigeminovascular system activation, followed by dural neurogenic inflammation and sensitization phenomenon, seems to be the main mechanisms in the migraine attacks. There are different types of medicines in the treatment of acute migraine attacks, such as triptans, ergotamines and analgesics [2,3]. However, these agents often have an insufficient efficacy and are accompanied with a numerous side effects [4,5,6]. The effective management of migraine remains a moving field and needs to be sharply into focus. There is a growing body of evidence supporting the efficacy of complementary and alternative medicine approaches in the management of migraine. Many herbs and herbal formulas are effective for migraine prevention and treatment [7]. In particular, Chuanxiong Rhizoma and Cyperi Rhizoma (CRCR) have been utilized together to cure migraine in many formulae [8]. These formulae usually contain a variety of herbs, among which CRCR are considered as important herbs. However, the effect of the combination of CRCR on migraine remains unclear.

Surprisingly, CRCR, an ancient and classic formula comprised of Chuanxiong Rhizoma and Cyperi Rhizoma in a weight ratio of 1:2, has been used to treat migraine dating back to the Danxi Xinfa treatise published during the Yuan Dynasty (AD 1347) of China. Chuanxiong Rhizoma (rhizomes of *Ligusticum chuanxiong* Hort.), has long been used for curing migraine, cardiovascular and cerebrovascular diseases [9]. Phytochemical studies identified the active components as phthalide lactones and phenolic acids, including ferulic acid (FA), senkyunolide A (SA), 3-*n*-butylphthalide (NBP), *Z*-ligustilide (LIG), *Z*-3-butylidenephthalide (BDPH) and so on [10,11]. Cyperi Rhizoma (rhizomes of *Cyperus rotundus* L.) displays anticonvulsant, anti-inflammatory, analgesic, antioxidant and antidepressant properties [12,13]. The most abundant ingredients are sesquiterpene lactones, among which are cyperotundone (CYT), nookatone (NKT), α-cyperone (CYP), cyperene and rotundene [14,15]. CYT, NKT and CYP were found in relative high levels compared with other compounds. According to the traditional Chinese medicine theory and modern scientific studies, we presume that CRCR may have tremendous advantages in the treatment of migraine. Because CRCR contains just two herbs, both of them can be used in health foods, which further confirms their safety. It can be supposed that CRCR may be beneficial as a combinational and complementary phytomedicine for migraine treatment. However, CRCR are short of experimental studies on pharmacological actions and pharmacodynamic material basis. Li reported that a UPLC-MS/MS method was developed for simultaneous quantification of phthalides and aromatic acids in Chuanxiong Rhizoma extract [16]. FA, SA, NBP, LIG, BDPH and levistilide A in rat plasma were simultaneously quantified by a UPLC-MS/MS method [17,18]. Wang reported that CYT, NKT and CYP were simultaneously quantified in Cyperi Rhizoma herb by a HPLC method [19]. However, there are no researches on quantification of CYT, NKT and CYP in biological samples. This study aimed to explore the anti-migraine effect and - active constituents in CRCR. Firstly, the changes of cerebral blood flow were detected by laser Doppler perfusion monitoring. Migraine-related vasoactive substances, neurotransmitters and neuropeptides, including endothelin-1 (ET-1), nitric oxide synthase (NOS), nitric oxide (NO), 5-hydroxytryptamine (5-HT), 5-hydoxyindoleacetic acid (5-HIAA), γ-aminobutyric acid (GABA), calcitonin gene-related peptide (CGRP) and β-endorphin (β-EP), were observed with enzyme-linked immunosorbent assay (ELISA). CGRP and c-Fos mRNA expression were quantified with quantitative real-time polymerase chain reaction (qRT-PCR). All these methods were used to evaluate the therapeutic effects on nitroglycerin (NTG)-induced migraine rats. Besides, a UPLC-MS/MS method was firstly developed to simultaneously quantify the main active constituents of CRCR in rat serum and cerebral cortex, including FA, SA, NBP, LIG, BDPH, CYT, NKT and CYP. Both phytochemical and pharmacological effects of CRCR on the migraine treatment were evaluated in this paper.

## 2. Results

### 2.1. Effects of CRCR on Cerebral Blood Flow

The cerebral blood flow was measured 2 h after inducing migraine model by a laser Doppler perfusion monitor (Figure 1A). As shown in Figure 1B, the cerebral blood flow significantly decreased in the NTG group compared to the control group (*p* < 0.01). CRCR 4.4 g/kg and 6.6 g/kg prominently improved the cerebral blood flow compared to the NTG group (*p* < 0.01 or 0.01 vs. model). It can be concluded that the cerebral blood flow was ameliorated by CRCR treatment.

### 2.2. ET-1, GABA, NOS and NO Levels in the Serum

The levels of ET-1, GABA, NOS and NO in the serum in migraine model rats were significantly higher than those in the control rats (*p* < 0.01, *p* < 0.05, *p* < 0.01, *p* < 0.01). ET-1 and NOS levels in the sumatriptan-treated migraine rats, were significantly lower than those in the migraine model rats (*p* < 0.01, *p* < 0.01). Treatment with CRCR 4.4 g/kg significantly reduced ET-1 and NOS levels compared to the NTG-induced migraine model rats (*p* < 0.05, *p* < 0.05). The levels of ET-1, GABA and NOS were remarkably down-regulated with the treatment of CRCR 6.6 g/kg (*p* < 0.05, *p* < 0.05, *p* < 0.01). Data are shown in Figure 2A.

### 2.3. 5-HT, 5-HIAA, CGRP and β-EP Levels in the BrainStem

Compared with normal rats, the levels of 5-HT, 5-HIAA and β-EP in the brainstem significantly decreased (*p* < 0.05, *p* < 0.05, *p* < 0.01), while CGRP level significantly increased (*p* < 0.05) in the NTG-induced migraine rats. Treatment with sumatriptan significantly increased 5-HT, 5-HIAA and β-EP levels (*p* < 0.01, *p* < 0.05, *p* < 0.05) and decreased CGRP level (*p* < 0.01). CRCR 4.4 g/kg administration caused great increases in 5-HIAA and β-EP levels (*p* < 0.05, *p* < 0.05), and a decrease in CGRP (*p* < 0.01). CRCR 6.6 g/kg significantly increased the levels of 5-HT, 5-HIAA and β-EP (*p* < 0.01, *p* < 0.01, *p* < 0.05), and decreased CGRP level (*p* < 0.01). Data are shown in Figure 2B.

### 2.4. mRNA Expressions of CGRP and c-Fos in the Brainstem

NTG injection caused marked increases in CGRP and c-Fos mRNA expressions compared to the control group (*p* < 0.05, *p* < 0.01). Sumatriptan inhibited CGRP and c-Fos mRNA expressions compared to the NTG group (*p* < 0.05, *p* < 0.01). Meanwhile, significant decreases in CGRP and c-Fos mRNA levels were observed in the migraine rats after oral administration of 4.4 g/kg and 6.6 g/kg CRCR. Consequently, CRCR prevented the increase of CGRP and c-Fos mRNA levels in the NTG-induced migraine rats (Figure 3).

### 2.5. Optimization for UPLC-MS/MS Parameters

During the optimization of MS conditions, the precursor and product ions of analytes were investigated by using the standard solution in the multiple reaction monitoring modes. The analytes showed different response values in the positive and negative ion mode. Ionization of SA, NBP, LIG, BDPH, CYT, NKT and CYP yielded much higher abundance in the positive mode. However, the ionization of FA showed much stronger intensities in the negative ionization mode. Finally, a method involving both positive and negative ionization modes was selected to allow for simultaneous detection. The MS parameters, such as nozzle voltage, capillary voltage, the flow rate of ion source gas, dwell and collision energy, were also optimized to obtain the highest intensity.

To optimize the chromatographic conditions, various mobile phase solvents and addtives, including methanol, acetonitrile, formic acid, acetic acid, were assessed to obtain symmetric peak shapes and short retention times for all analytes. Finally, the mobile phase consisting of methanol and water with 0.1% formic acid was utilized in a gradient elution program to obtain satisfactory peak symmetry, good resolution and significantly enhanced sensitivity. 

A key factor in biological analysis is to select a suitable internal standard (IS), which can rectify the probable error in sample determination. Usually an isotope-labeled internal standard is considered ideal [20]; however, it is very expensive and not easy to obtain such compounds. In this study, buspirone (BUS) and genistein (GEN) were selected as the positive-ion and negation-ion internal standard, respectively. Both them showed good separation from the analytes and no endogenous existing in the matrix.

### 2.6. UPLC-MS/MS Method Validation

#### 2.6.1. Selectivity

Under the optimized LC-MS conditions, the representative chromatograms of blank serum, blank serum spiked with the analytes, and rat serum sample after oral CRCR are shown in Figure 4.

The chromatograms of blank cerebral cortex homogenate, spiked cerebral cortex homogenate with standards, and rat cerebral cortex homogenate after oral CRCR are shown in Figure 5. No significant interferences from endogenous matrix components were observed in the retention time of the analytes and internal standards, suggesting good selectivity of the developed method.

#### 2.6.2. Linearity and LLOQ

The regression equations for FA, SA, NBP, LIG, BDPH, CYT, NKT and CYP in serum were: Y = 0.096X + 0.072 (R = 0.999) for FA; Y = 0.059X + 0.133 (R = 0.995) for SA; Y = 0.087X + 1.247 (R = 0.998) for NBP; Y = 0.027X + 0.106 (R = 0.999) for LIG; Y = 0.043X + 0.065 (R = 0.998) for BDPH; Y = 0.066X + 0.596 (R = 0.999) for CYT; Y = 0.052X + 0.219 (R = 0.999) for NKT; Y = 0.096X + 0.702 (R = 0.999) for CYP. The linearity ranges of FA, SA, NBP, LIG, BDPH, CYT, NKT and CYP in serum were 8–1000, 8–1000, 5–1000, 5–1000, 5–1000, 5–1000, 5–1000, and 5–1000 ng/mL, respectively. The LLOQ of FA, SA, NBP, LIG, BDPH, CYT, NKT and CYP in serum were 8, 8, 5, 5, 5, 5, 5, 5 ng/mL, respectively. The RSDs at the LLOQ were 4.20–19.48%, and REs at the LLOQ were −4.65–13.74%, which were acceptable with an RSD of 20% and an RE within ±15%.

The regression equations for SA, NBP, LIG, BDPH, CYT and CYP in the cerebral cortex homogenate were: Y = 0.150X + 0.031 (R = 0.994) for SA; Y = 0.280X + 0.140 (R = 0.994) for NBP; Y = 0.036X + 0.049 (R = 0.994) for LIG; Y = 0.029X − 0.010 (R = 0.993) for BDPH; Y = 0.246X + 0.491 (R = 0.993) for CYT; Y = 0.438X + 0.872 (R = 0.981) for CYP. The linear regressions exhibited good linear relationships over the range of 5–1000 ng/mL. The LLOQ was 5 ng/mL for all the six analytes. The RSD and RE of the six analytes at the LLOQ were less than 18.64% and within ±8.57%.

#### 2.6.3. Precision and Accuracy

As shown in Table 1, the results for intra- and inter-day precisions and accuracies for the analytes in serum indicated that the intra- and inter-day RSDs were 0.95–15.38% and 0.58–10.77%, respectively, while the corresponding REs were from −15.84% to 15.81% and from −13.98% to 13.95%, respectively. Table 2 summarizes the intra- and inter-day precisions and accuracies of the analytes in the cerebral cortex homogenate. The intra- and inter-day RSDs were all less than 15%, while the corresponding REs were within the range of ±15%. These results suggested that the precision and accuracy were in the acceptable ranges, and the developed method proved to be precise and accurate 

#### 2.6.4. Matrix Effect and Extraction Recovery

The matrix effects were determined in three replicates at low (10 ng/mL), medium (100 ng/mL) and high (800 ng/mL) concentrations of analytes. The matrix effect values ranged 93.90–116.61% for the analytes in serum (Table 3), and ranged 86.62–103.36% for the analytes in the cerebral cortex homogenate (Table 4), suggesting that ion suppression or ion enhancement from matrix was negligible in this current method. The recoveries of the analytes in serum were found to be 85.29–108.49%, with RSD of less than 6.09% (Table 3). The recoveries of the analytes in the cerebral cortex homogenate were between 84.96% and 107.23% with RSD were lower than 15% (Table 4). All of these values indicated acceptable matrix effect and recovery for the present method.

#### 2.6.5. Stability

The results of short-term stability, long-term stability and freeze-thaw stability of the analytes in serum were summarized in Table 5, which showed that RSDs were 0.30–8.83% and REs were from −15.99% to 14.18%. As shown in Table 6, the results for the analytes in the cerebral cortex homogenate indicated that RSDs were 0.30–12.59% and REs were from −16.80% to 18.76%. Thus, the analytes were stable under these storage conditions.

### 2.7. The Concentrations of Active Constituents in Rat Serum and Cerebral Cortex Determined by the Developed UPLC-MS/MS Method

The dynamic curves for FA, SA, NBP, LIG, BDPH, CYT, NKT and CYP in the CRCR extract, rat serum and rat cerebral cortex are presented in Figure 6. The concentrations of the analytes in 6.6 g/kg CRCR extract were higher than those in the 4.4 g/kg CRCR extract. The contents of the analytes in rat serum and cerebral cortex in NTG + CRCR 6.6 g/kg group were much higher than those in NTG + CRCR 4.4 g/kg group. An obvious trend can be seen that the higher drug concentrations there are in the CRCR extract, the higher drug levels there can be qualified in the rat serum and cerebral cortex. Secondly, the analytes contents in rat serum and cerebral cortex in NTG + CRCR 4.4 g/kg group were much higher than those in CRCR 4.4 g/kg. It can be concluded that NTG, as a superior vasodilator, could promote drug absorption into blood. Thirdly, FA, SA, NBP, LIG, BDPH, CYT, NKT and CYP were qualified in rat serum and cerebral cortex, while the contents of FA and NKT were too low to be detected in the cerebral cortex. SA, NBP, LIG, BDPH and CYT seemed to be the leading components in serum. FA, NKT and CYP had lower concentrations in the serum. The most abundant compound in the cerebral cortex was SA and this was followed by LIG and NBP. The contents of BDPH, CYT and CYP in the cerebral cortex were not high.

## 3. Discussion

Migraine is a chronic, complex neurovascular disorder characterized by multiphase attacks of head pain and a myriad of neurological symptom. The pathogenesis of migraine is very complex, in which both central and peripheral trigeminal pain pathways probably play a significant role. NTG-induced migraine model and cortical spreading depression migraine model are two classical migraine models. The administration of NTG, an NO donor, provokes spontaneous migraine-like attacks [21]. In this study, we focused on the role of CRCR on treating NTG-induced migraine models, and other animal models of migraine need to be further explored.

Given the side effects of anti-migraine medications, there is an increasing demand for herbal treatment. In recent years, combinational herbal therapy for migraine has been proposed to further improve effects [22,23]. This study indicated that CRCR exerted powerfully preventive and therapeutic effects on migraine via increasing the cerebral blood flow, decreasing the expression of CGRP and c-Fos mRNA, and regulating the releasing of ET-1, GABA, NOS, 5-HT, 5-HIAA, CGRP and β-EP in the serum and brainstem, consequently relieving neurogenic inflammation.

During the different phases of the migraine attack, there are variations in cerebral blood flow, coinciding with changes in cerebrovascular activity. Kastrup reported increased cerebral blood flow in migraine patients [24]. Frieberg reported reduced cerebral blood flow in middle cerebral artery on the headache side during a migraine attack when compared to the non-headache side and which was acquired during an attack-free episode [25]. In our study, Laser Doppler perfusion monitor was used to measure cerebral blood flow. It can be noted that the cerebral blood flow significantly decreased in the NTG-induced migraine rats. CRCR treatment increased the cerebral blood flow. These findings demonstrated that treatment with CRCR improved imbalance of cerebral blood flow during migraine.

As we know, cerebral vascular dysfunction is always induced by abnormal levels of vasoactive substances, such as ET-1, NO and NOS. Recent studies have highlighted a potentially important role for the biological endothelial function in the migraine pathophysiology [26]. It involved the most potent endothelial-derived constricting factor, ET-1, and its endothelial counterpart, NO, in the vascular changes during migraine attacks [27]. ET-1 can induce cortical spreading depression in the attack-triggering cascade of migraine attacks with and without aura [28,29]. Elevated ET-1 plasma levels have been observed in the early phase of migraine attacks [30,31]. NO is a small gaseous signaling molecule, and is very important in migraine. NO is endogenously produced by NOS. A number of NO signaling pathways have been shown to be up-regulated in migraine patients [32,33]. 

It has been demonstrated that some neurotransmitters play a crucial role in migraine [34]. 5-HT is widely accepted as a contributor to migraine. 5-HT releases in the early prodromal phase of a migraine attack, subsequently, 5-HT is metabolized to 5-HIAA in the ictal phase [35]. Migraine patients have a low plasma and cerebral 5-HT level between attacks [36]. GABA is an inhibitory neurotransmitter in the central nervous system. GABA has been implicated in clinical conditions thought to involve regulating of the balance between excitatory and inhibitory process [37]. Increased GABA levels have been reported during migraine [38].

Neurogenic inflammation resulting from neuropeptides is important for inducing and aggravating migraines. CGRP, as the strongest vasodilator, is released from trigeminal neurons and dural tissue, which can stimulate meningeal vasodilatation, activate peripheral meningeal nociceptors causing migraine [39,40]. β-EP is beneficial to migraine through inhibition of neuronal firing [41]. Clinical studies showed decreased β-EP concentrations in migraine patients [42], and low β-EP levels have reflected low analgesic activity.

Based on the above views, we found that CRCR could remarkably decrease the levels of ET-1, GABA and NOS in serum in the migraine rats. Treatment with CRCR significantly increased the levels of 5-HT, 5-HIAA and β-EP in the brainstem in the migraine rats. Additionally, both CGRP and CGRP mRNA expression in the brainstem were markedly down-regulated with the treatment of CRCR. c-Fos expression is commonly used as a biological marker of painful stimuli [43]. Here, we showed that NTG injection induced an increase in c-Fos mRNA in the brainstem, while CRCR could significantly decrease c-Fos mRNA expression. These above findings demonstrated that CRCR could exert analgesic effect on NTG-induced migraine through adjusting in the levels of relevant vasoactive substances, neurotransmitters and neuropeptides.

Next, the anti-migraine active constituents were investigated. The main components in CRCR extract, rat serum and cerebral cortex were qualified, respectively. From CRCR extract to blood, and then from blood to brain tissues, active constituents were determined step by step. FA, SA, NBP, LIG, BDPH, CYT, NKT and CYP were found in CRCR extract and rat serum. SA, NBP, LIG, BDPH, CYT and CYP were discovered in rat cerebral cortex, while the contents of FA and NKT were too low to be detected in the cerebral cortex. Migraine is widely believed to be associated with neurogenic inflammation. As reported, FA, SA, NBP, LIG, BDPH, CYT, NKT and CYP displayed anti-inflammatory and neuroprotective activity [44,45,46,47,48]. Accordingly, we supposed that active constituents in CRCR for treating migraine were FA, SA, NBP, LIG, BDPH, CYT, NKT and CYP.

## 4. Materials and Methods

### 4.1. General Information

Chuanxiong Rhizoma and Cyperi Rhizoma were obtained from Sichuan Neautus Traditional Chinese Co., Ltd. (Chengdu, China)-. Buspirone (positive internal standard, +IS, CAS 36505-84-7) was purchased from Sigma-Aldrich (St Louis, MO, USA). Genistein (negative internal standard, -IS, CAS 446-72-0), ferulic acid (CAS 1135-24-6) and 3-*n*-butylphthalide (CAS 6066-49-5) were obtained from the National Institutes for Food and Drug Control (Beijing, China). Senkyunolide A (CAS 63038-10-8), *Z*-ligustilide (CAS 81944-09-4), *Z*-3-butylidenephthalide (CAS 72917-31-8), cyperotundone (CAS 3466-15-7), nookatone (CAS 4674-50-4) and α-cyperone (CAS 473-08-5) were purchased from PUSH Bio-Technology (Chengdu, China). All the chemicals were with a purity of greater than 98%, and their chemical structures are shown in Figure 7. Nitroglycerin injections (batch No. 20170808) were obtained from Beijing Yimin Pharmaceutical Co., Ltd. (Beijing, China-). Sumatriptan succinate tables (batch No. 7B6926T) were purchased from Tianjin Huajin Pharmaceutical Co., Ltd. (Tianjin, China-). Methanol (HPLC grade) was obtained from Fisher Scientific Inc (Fairlawn, NJ, USA). Formic acid was purchased from Sigma-Aldrich. Ultrapure water was produced by a Milli-Q Reagent Water System (Millipore, MA, USA). All other chemicals were of analytical grade.

### 4.2. Preparation and Standardisation of CRCR Extract

Chuanxiong Rhizoma and Cyperi Rhizoma were extracted twice with 70% ethanol (1:6, w/v) for 1 h per time. The extraction solutions were combined and evaporated under vacuum to remove ethanol. The solutions were further concentrated on a water bath, and then dried in vacuum. The produced power was dissolved in the water for intragastric administration. The contents of the main ingredients in the CRCR extract were determined by the established HPLC method [49]. For HPLC analysis, 1.5235 g powder was accurately weighed and extracted with 25 mL methanol by ultrasonic for 20 min. The weight loss in the ultrasonic procedure was compensated, and then filtered through 0.45 µm membrane filter prior injection into the HPLC system. The concentrations of the main ingredients in the CRCR extract were shown in Figure 6. The HPLC chromatograms of CRCR extract solution were shown in Appendix A.

### 4.3. Animals

Adult male Sprague-Dawley rats (weight 180–220 g) were purchased from Beijing Vital River Laboratory Animal Technology Co., Ltd. (Beijing, China). Animals were approved by the Animal Ethics Committee of Capital Medical University (Beijing, China). All experimental procedures were approved by the Experimental Animal Care and Use Committee of Capital Medical University (Beijing, China). Rats were housed in an air-conditioned room at a temperature of 23 ± 2 °C, with a relative humidity of 55 ± 10% with water and food available ad libitum. Rats were acclimated in the laboratory for at least 3 days prior to the experiment and fasted for 12 h but allowed water ad libitum before experiments.

### 4.4. Experimental Plan

Rats were randomly allocated into six groups (n = 9/group): control (saline, 10 mL/kg), NTG (saline, 10 mL/kg), CRCR 4.4 g/kg (CRCR, 4.4 g/kg), NTG + sumatriptan (sumatriptan, 5.83 mg/kg), NTG+CRCR 4.4 g/kg (CRCR, 4.4 g/kg) and NTG+CRCR 6.6 g/kg (CRCR, 6.6 g/kg). The dosage of sumatriptan succinate tables was calculated according to the single oral dose used clinically in adults [50]. Rats were subjected to intragastric administration of corresponding drugs for consecutive five days. Rats were subcutaneously injected with 10 mg/kg NTG 30 min after the last treatment, except in the control group and CRCR 4.4 g/kg group. 2 h after NTG injection, rats were anesthetized with 10% chloral hydrate (3.5 mL/kg). The whole skull was exposed to detect the cerebral blood flow. And then blood samples were collected via the abdominal aorta. Serum was separated via centrifugation at 3000 rpm for 10 min at 4 °C after standing for 2 h, and the serum was frozen at −80 °C until analysis. After collecting blood, rats were sacrificed, and brainstem and cerebral cortex were separated on ice. Brainstem was divided into left and right lobes along the midline, and processed either for the quantification of 5-HT, 5-HIAA, CGRP and β-EP with ELISA or for the detection of CGRP and c-Fos mRNA expression with qRT-PCR. ET-1, GABA, NOS and NO levels in serum were also quantified using ELISA. The concentrations of FA, SA, NBP, LIG, BDPH, CYT, NKT and CYP, as the main ingredients in CRCR, were simultaneously quantified in the rat serum and cerebral cortex, by the developed UPLC-MS/MS method.

### 4.5. Cerebral Blood Flow Measurement

A laser Doppler perfusion monitor (PeriCam PSI System, Perimed, Stockholm, Sweden) was used to measure cerebral blood flow. A crossing skin incision was made on the head of rat to expose the whole skull under anesthesia. Laser scanning imaging measurements were performed on the intact skull. The laser beam was pointed to the bregma, the scanning distance was set up as 15.0 cm and the scanning range was 2.0 cm × 2.0 cm. A round window of approximately 0.8 mm^2^ in size had been located right and down of the bregma (2 mm posterior, 5 mm lateral to bregma). The frequency required to measure the blood flow and generate the image was 5 images per second. The mean cerebral blood flow was calculated by the PeriCam PSI system.

### 4.6. ELISA

The levels of 5-HT, 5-HIAA, CGRP and β-EP in the brainstem tissue and ET-1, GABA, NO and NOS in the serum were determined by ELISA using the rat 5-HT, 5-HIAA, CGRP, β-EP, ET-1, GABA, NO and NOS ELISA kits (Nanjing Jiancheng Bioengineering Institute, -Nanjing, China). Briefly, the left brainstem tissues were homogenized in 1:9 (wt/vol) 0.1 M PBS solutions on ice, and centrifuged at 3000 r/min for 20 min at 4 °C. Supernatants were used immediately or directly frozen, and stored at −80 °C. Serum was separated through centrifugation at 3000 rpm/min for 10 min at 4 °C. Each multiplex assay was performed in accordance with the manufacturer’s instructions. Absorbance was measured in the Multiskan Go Microplate Spectrophotometer (Thermo Fisher Scientific, Inc. Waltham, MA, USA).

### 4.7. qRT-PCR

Both CGRP and c-Fos mRNA expression levels in brainstem tissue were detected by qRT-PCR as follows: Total RNA was extracted from the right brainstem tissue using TRIZOL reagent (Invitrogen, Carlsbad, CA, USA) according to the manufacturer’s instructions. Single-stranded cDNA was synthesized from 1 µg RNA using reverse transcription kit (Tiangen Biotech Co., Ltd., Beijing, China). The amplification was performed by SuperReal PreMix Plus (SYBR Green) kit (Tiangen Biotech Co., Ltd.) in the CFX Connect Real-time PCR Detection System (Bio-Rad, Hercules, CA, USA). The PCR protocol was as follows: predenaturation for 15 min at 95 °C followed by denaturation for 10 s at 95 °C, annealing for 20 s at 52 °C, and extension for 30 s at 72 °C for 40 cycles. Primers for gene amplification were as follows: CGRP forward (5′-CAGTCTCAGCTCCAAGTCATC-3′, Tm = 57.57 °C) and CGRP reverse (5′-TTCCAAGGTTGACCTCAAAG-3′, Tm = 53.35 °C); c-Fos forward (5′-TACG CTCCAAGCGGAGAC-3′, Tm = 57.18 °C) and c-Fos reverse (5′-TTTCCTTCTCTTTCAGTAGAT TGG-3′, Tm = 54.44 °C); GAPDH forward (5′-AACCTGCCAAGTATGATGAC-3′, Tm = 53.35 °C) and GAPDH reverse (5′-GGAGTTGCTGTTGAAGTCA-3′, Tm = 53.01 °C). The CGRP and c-Fos mRNA expression levels were normalized to GAPDH. All relative expression levels were calculated through 2^−△△Ct^ method. The specificity of these amplifications was verified by melt curve analysis with detection of only a single peak.

### 4.8. Chromatographic and Mass Spectrometric Conditions

Chromatographic analyses were performed using an Agilent 1290 series UPLC system (Agilent Technologies, Santa Clara, CA, USA) equipped with a G4220A pump, a G1379A online degasser, a G4226 autosampler and a G1330B thermostatically controlled column compartment. The separations were achieved on an Agilent InfinityLab Poroshell 120 EC-C18 column (100 mm × 2.1 mm, 2.7 µm) with the column temperature set at 35 °C. The mobile phases consisted of 0.1% formic acid (A) and methanol (B) using a gradient elution as follows: 0.00 min 20% B, 1.00 min 20% B, 4.00 min 60% B, 10.10 min 70% B, 12.00 min 95% B, 13.00 min 95% B, 14.00 min 100% B, with the flow rate of 0.3mL/min. The injection volume was set at 3 µL and the running time was 14 min. 

The MS detection was performed using an Agilent 6490 triple quadrupole mass spectrometer (Agilent Technologies) with a Jetstream electrospray source in both positive and negative ionization mode. Quantification was performed using multiple reaction monitoring of the precursor product ion transition at m/z 192.8→133.8 for FA with a collision energy of −10 eV; m/z 193.0→137.0 for SA with a collision energy of +13 eV; m/z 190.9→145.0 for NBP with a collision energy of +16 eV; m/z 191.0→90.9 for LIG with a collision energy of +29 eV; m/z 189.0→128.0 for BDPH with a collision energy of +20 eV; m/z 219.0→109.0 for CYT with a collision energy of +26 eV; m/z 219.0→81.0 for NKT with a collision energy of +23 eV; m/z 219.0→111.0 for CYP with a collision energy of +20 eV; m/z 386.0→149.9 for BUS with a collision energy of +23 eV and m/z 269.7→132.8 for GEN with a collision energy of −40 eV. The other parameters of the mass spectrometer were as follows: nozzle voltage, ±1.5 kV; capillary voltage, ±3.0 kV; nebulizer 20 psi; sheath gas flow, 11.0 L/min nitrogen; sheath gas temperature 250 °C; gas flow, 14.0 L/min nitrogen; gas temperature 200 °C. The collision gas for MS/MS was high purity nitrogen for collision-induced dissociation. MassHunter workstation B.0600 software was used for quantitative analysis (Agilent Technologies, Santa Clara, CA, USA).

### 4.9. Preparation of the Stock and Working Solutions

The stock solutions (1 mg/mL) of FA, SA, NBP, LIG, BDPH, CYT, NKT, CYP, BUS and GEN were prepared by dissolving the accurately weighed reference compounds in methanol. Stock solutions of FA, SA, NBP, LIG, BDPH, CYT, NKT and CYP were then mixed and serially diluted with methanol to obtain standard working solutions containing 1.0–1000 ng/ml FA, SA, NBP, LIG, BDPH, CYT, NKT and CYP. The IS solution (20 ng/mL BUS and 100 ng/mL GEN) was prepared by diluting the stock solutions (1 mg/mL) with methanol: acetonitrile (50:50, *v*/*v*). All the stock and working solutions were stored at 4 °C prior to use.

### 4.10. Preparation of Calibration Standards and Quality Control Samples

Calibration curves, low, medium and high concentrations of quality control (QC) samples were prepared by spiking 100 µL of standard working solutions and 20 µL of IS solution (20 ng/mL BUS and 100 ng/mL GEN) to 100 µL of blank rat serum or blank rat cerebral cortex homogenate (drug-free). Calibration standards were prepared at concentrations of 1.0, 5.0, 10.0, 25.0, 50.0, 100, 200, 400, 500, 800 and 1000 ng/mL of FA, SA, NBP, LIG, BDPH, CYT, NKT and CYP, while QC samples were prepared at 10.0, 100 and 800 ng/mL of FA, SA, NBP, LIG, BDPH, CYT, NKT and CYP to evaluate the precision, accuracy, stability, matrix effect and recovery of analytical method.

### 4.11. Serum Sample and Cerebral Cortex Sample Treatment for UPLC-MS/MS

Aliquots of 100 µL serum were mixed with 100 µL methanol (or standard working solution or QC solution), 20 µL of IS solution (20 ng/mL BUS and 100 ng/mL GEN) and 300 µL of methanol: acetonitrile (50:50, *v*/*v*).

The cerebral cortex was homogenized in 1:1 (wt/vol) ice-cold water on ice. Aliquots of 400 µL cerebral cortex homogenate were mixed with 100 µL methanol (or standard working solution or QC solution), 20 µL of IS solution (20 ng/mL BUS and 100 ng/mL GEN) and 2 mL of methanol: acetonitrile (50:50, *v*/*v*).

After vortexing for 1 min and then centrifuging at 9500× *g* for 10 min, the supernatant was transferred to a clean Eppendorf tube and evaporated to dryness at 37 °C under a gentle stream of nitrogen. The residue was reconstituted with 100 µL methanol, vortex-mixed for 1 min, and centrifuged at 9500× *g* for 5 min. 3 µL of the supernatant was injected into the UPLC-MS/MS system for analysis.

### 4.12. UPLC-MS/MS Method Validation

The method was validated in terms of selectivity, linearity, lower limit of quantification (LLOQ), precision, accuracy, matrix effect, recovery and stability.

#### 4.12.1. Selectivity

To evaluate the selectivity, six samples of blank serum or blank cerebral cortex homogenate were analyzed by comparing with the corresponding spiked serum or spiked cerebral cortex homogenate samples.

#### 4.12.2. Linearity and LLOQ

The linearity was assessed via plotting the peak area ratio of analytes to IS versus the corresponding concentration of analytes with a weighting factor of 1/x^2^ by least squares linear regression. The LLOQ was determined in accordance to the baseline noise, considering a signal-to-noise ratio of 10:1.

#### 4.12.3. Precision and Accuracy

The precision and accuracy were determined via performing replicate analyses of QC samples spiked with low, medium and high concentrations against calibration standards. Six replicates of QC samples at each concentration level were evaluated on the same day (intra-day) and on three different day (inter-day). The precision was expressed by relative standard diviation (RSD%), and the accuracy was calculated as relative error (RE%).

#### 4.12.4. Matrix Effect and Extraction Recovery

In order to investigate whether the ionization of analytes could be influenced by the co-elution and endogenous substances, matrix effect was determined by comparing the analytes spiked after extraction with those of the corresponding standard solutions. The extraction recovery was investigated by comparing the analytes obtained from samples with the analytes spiked before extraction with those of the analytes spiked after extraction.

#### 4.12.5. Stability

The stability was assessed by analyzing QC samples at three concentrations under different sample storage conditions. Short-term stability was evaluated by analyzing the QC samples that were kept at room temperature for 24 h. Long-term stability was assessed after the QC samples had been stored at −80 °C for 30 days. Freeze-thaw stability was investigated after three consecutive freeze (−20 °C) and thaw (room temperature) cycles. All the stability testing QC samples were evaluated using the newly prepared calibration curves.

### 4.13. Statistical Analysis

Data were represented as mean values ± standard error (SE), and analyzed using SPSS 17.0 statistical software (SPSS Inc., Chicago, IL, USA). The statistical difference between groups was analyzed using one-way analysis of variance (ANOVA). A value of *p* < 0.05 was considered statistically significant.

## 5. Conclusions

The present study indicated that CRCR showed significantly therapeutic effects on migraine via increasing cerebral blood flow, decreasing the expression of CGRP and c-Fos mRNA, and regulating the releasing of ET-1, GABA, NOS, 5-HT, 5-HIAA, CGRP and β-EP in the serum and brainstem. Besides, a simple, sensitive and reliable UPLC-MS/MS method was developed for the simultaneous determination of FA, SA, NBP, LIG, BDPH, CYT, NKT and CYP in rat serum and cerebral cortex. Furthermore, this method was successfully applied to qualify the concentrations of these representative constituents in biological samples. It can be concluded that SA, NBP, LIG, BDPH, CYT and CYP had relatively high concentration both in the serum and cerebral cortex, which played a critical role in treating migraine. Current evidences supported that CRCR may be beneficial as a combinational and complementary phytomedicine for migraine treatment. The established method might also contribute to the assessment of safety and efficacy of CRCR in clinical practice.

## Figures and Tables

**Figure 1 molecules-24-02230-f001:**
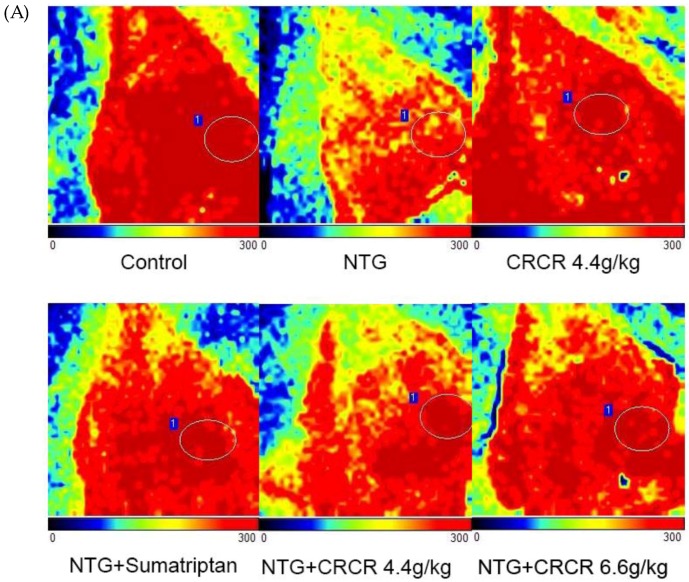
Effects of CRCR on cerebral blood flow. (**A**) Laser scanning images of cerebral blood flow in the cortex. (**B**) The mean cerebral blood flow values. Data are expressed as mean ± S.E. * *p* < 0.05, ** *p* < 0.01 vs. control group, ^#^
*p* < 0.05, ^##^
*p* < 0.01 vs. NTG group.

**Figure 2 molecules-24-02230-f002:**
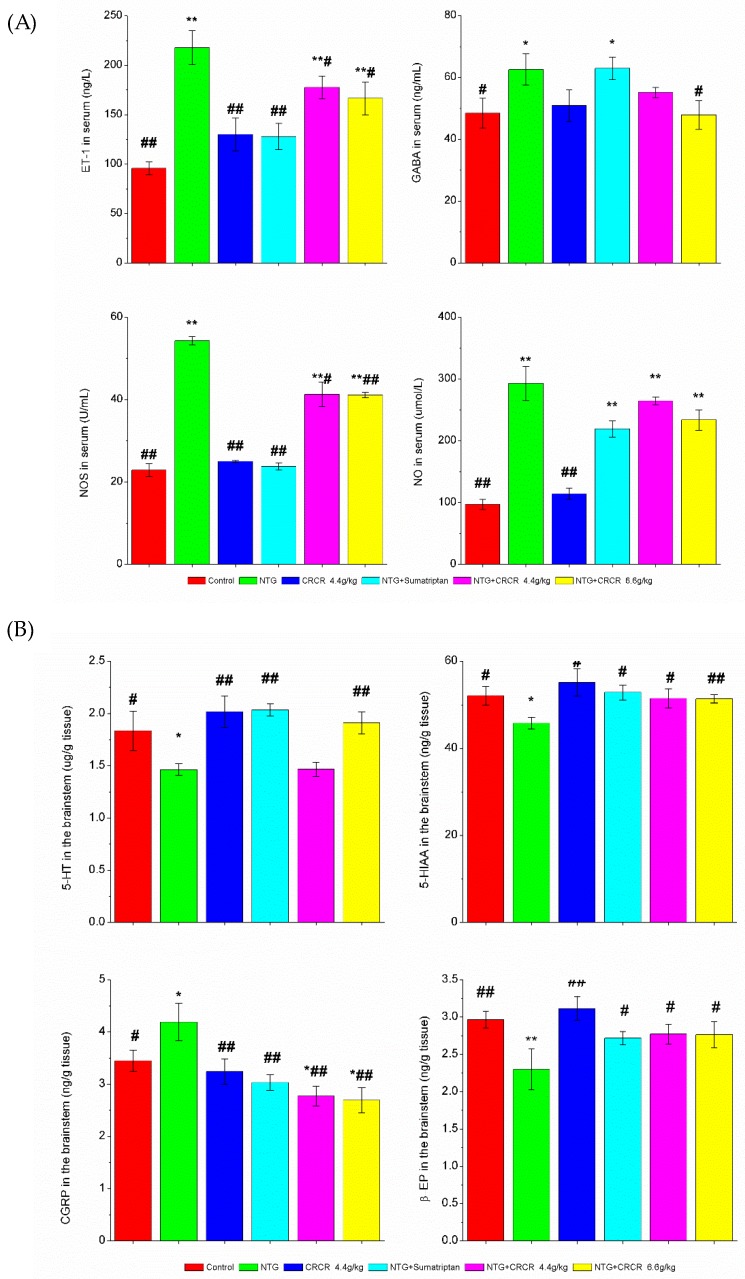
Effects of CRCR on ET-1, GABA, NOS, NO, 5-HT, 5-HIAA, CGRP and β-EP levels detected by ELISA. (**A**) ET-1, GABA, NOS and NO levels in the serum. (**B**) 5-HT, 5-HIAA, CGRP and β-EP levels in the brainstem. Data are expressed as mean ± S.E. * *p* < 0.05, ** *p* < 0.01 vs. control group, ^#^
*p* < 0.05, ^##^
*p* < 0.01 vs. NTG group.

**Figure 3 molecules-24-02230-f003:**
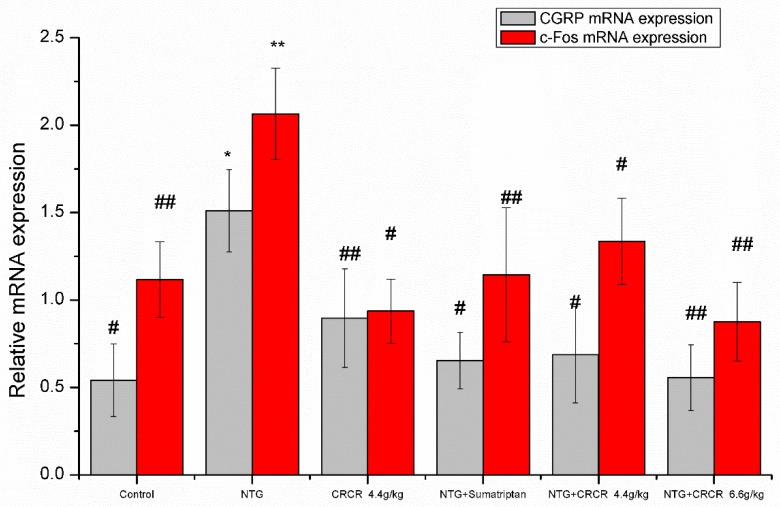
Effects of CRCR on CGRP mRNA and c-fos mRNA expression in the brainstem detected by qRT-PCR. Data are expressed as mean ± S.E. * *p* < 0.05, ** *p* < 0.01 vs. control group, ^#^
*p* < 0.05, ^##^
*p* < 0.01 vs. NTG group.

**Figure 4 molecules-24-02230-f004:**
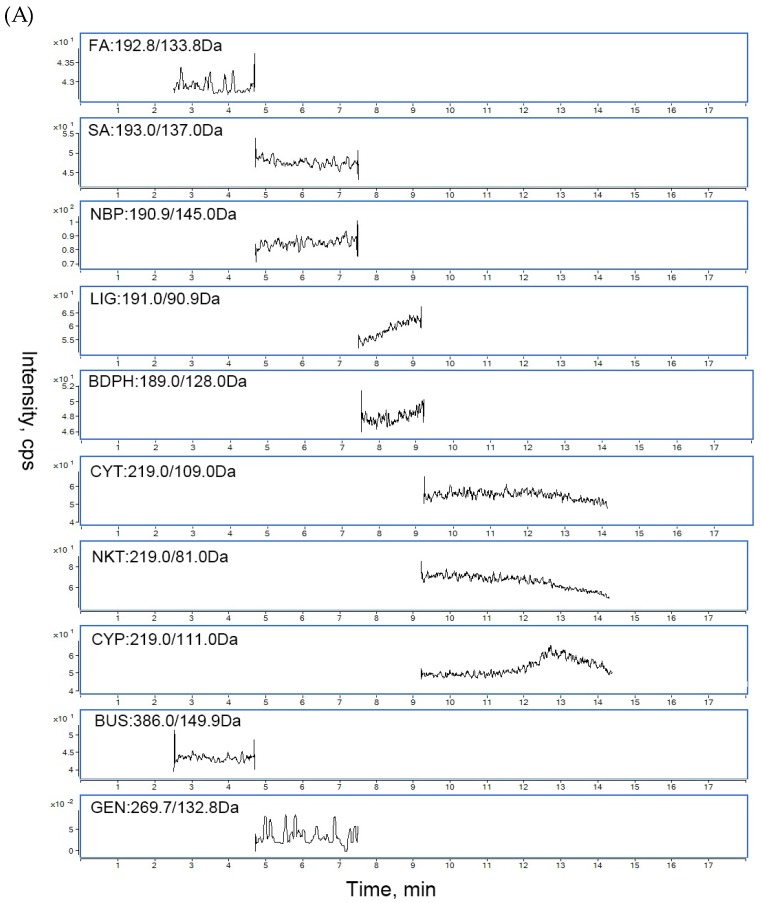
Typical chromatograms of (**A**) blank rat serum; (**B**) blank rat serum spiked with FA, SA, NBP, LIG, BDPH, CYT, NKT and CYP and IS; (**C**) an unknown rat serum sample after oral administration of 6.6 g/kg CRCR extract.

**Figure 5 molecules-24-02230-f005:**
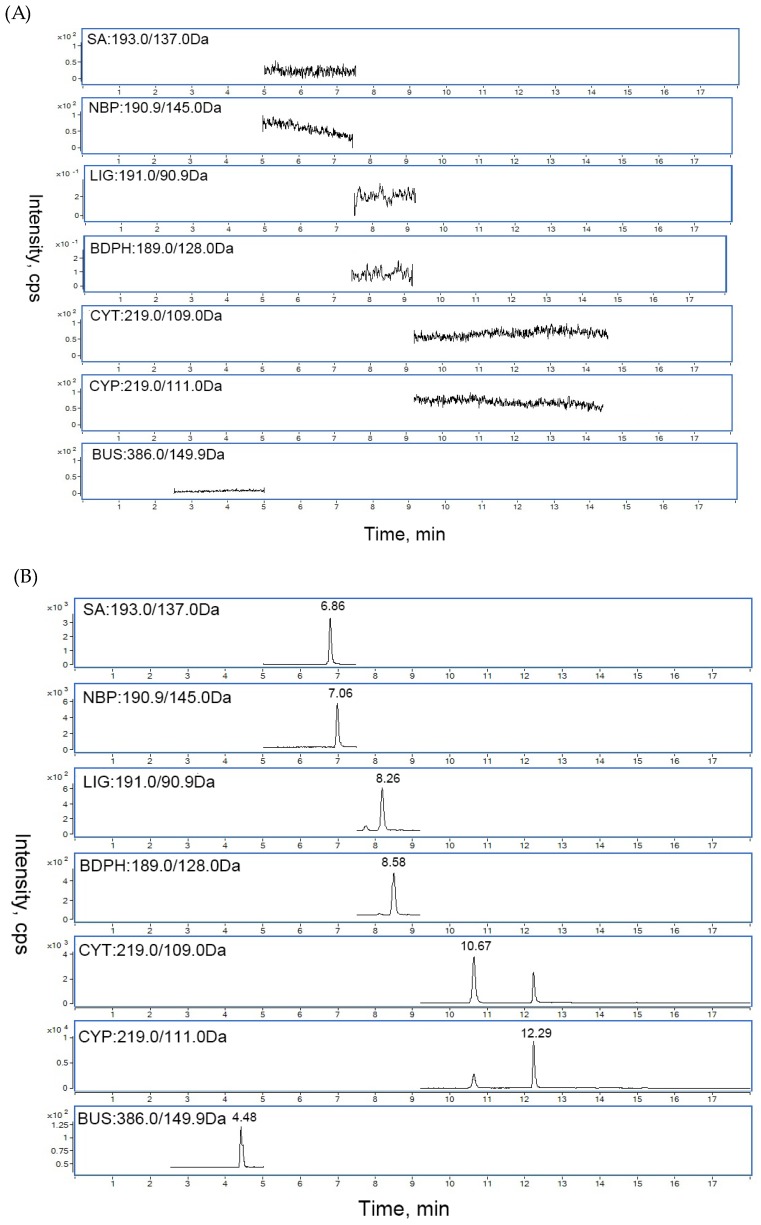
Typical chromatograms of (**A**) blank rat cerebral cortex homogenate; (**B**) blank rat cerebral cortex homogenate spiked with SA, NBP, LIG, BDPH, CYT and CYP and IS; (**C**) an unknown rat cerebral cortex homogenate sample after oral administration of 6.6 g/kg CRCR extract.

**Figure 6 molecules-24-02230-f006:**
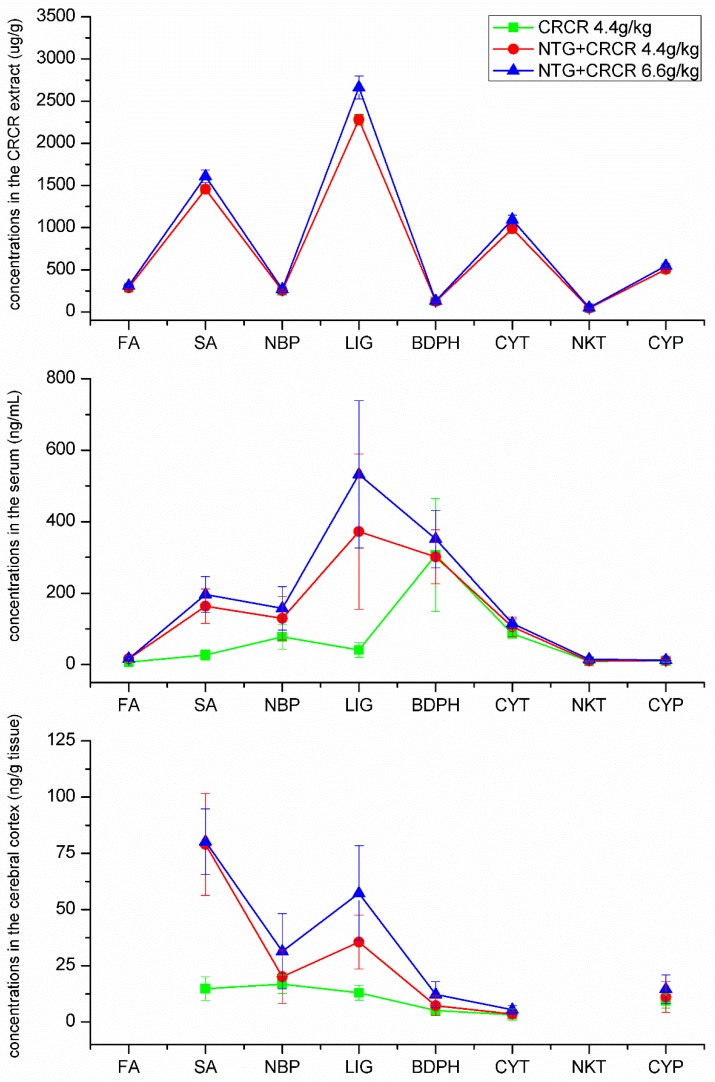
The concentrations of the analytes in the CRCR extract, rat serum and rat cerebral cortex determined by UPLC-MS/MS.

**Figure 7 molecules-24-02230-f007:**
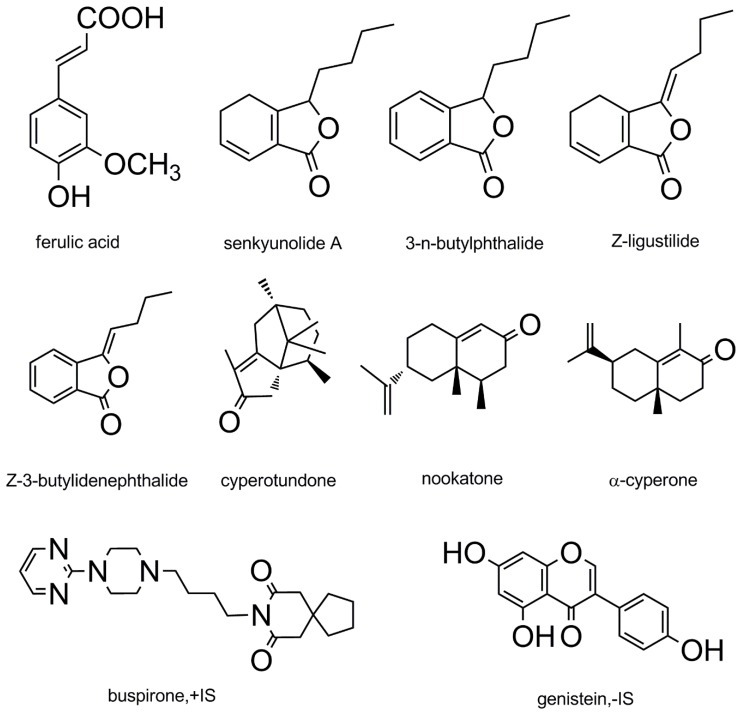
Chemical structures of FA, SA, NBP, LIG, BDPH, CYT, NKT, CYP, BUS and GEN.

**Table 1 molecules-24-02230-t001:** Intra-and inter-day precisions and accuracies of the analytes in rat serum (*n* = 6).

		Day 1			Day 2			Day 3			Inter-day		
Analytes	Spiked Conc. (ng/mL)	Mean ± SD (ng/mL)	RSD (%)	RE (%)	Mean ± SD (ng/mL)	RSD (%)	RE (%)	Mean ± SD (ng/mL)	RSD (%)	RE (%)	Mean ± SD (ng/mL)	RSD (%)	RE (%)
FA	10	9.01 ± 1.10	12.22	−9.87	9.69 ± 1.04	10.77	−3.09	8.56 ± 0.65	7.58	−14.37	9.01 ± 0.97	10.77	−9.91
	100	97.85 ± 8.02	8.19	−2.15	108.84 ± 5.10	4.68	8.84	97.86 ± 11.46	11.71	−2.14	101.52 ± 8.19	8.19	1.52
	800	743.52 ± 20.29	2.73	−7.06	−765.477 ± 33.65	4.40	−4.32	788.46 ± 26.90	3.41	−1.44	765.82 ± 26.94	3.51	−4.27
SA	10	10.33 ± 1.59	15.38	3.26	9.55 ± 1.38	14.47	−4.53	8.69 ± 0.86	9.90	−13.12	9.52 ± 0.40	4.20	−4.79
	100	100.99 ± 2.74	2.71	0.99	101.89 ± 3.07	3.01	1.89	103.35 ± 4.19	4.05	3.35	102.08 ± 3.38	3.31	2.08
	800	785.81 ± 38.86	4.95	−1.77	771.21 ± 7.51	0.97	−3.60	808.8 ± 15.66	1.94	1.10	788.61 ± 28.38	3.60	−1.42
NBP	10	9.96 ± 0.77	7.74	−0.42	9.08 ± 0.86	9.46	-9.20	9.22 ± 0.74	8.08	−7.78	9.42 ± 0.13	1.33	−5.80
	100	101.51 ± 2.98	2.93	1.51	101.24 ± 4.76	4.71	1.24	102.32 ± 5.97	5.83	2.32	101.69 ± 4.53	4.45	1.69
	800	809.29 ± 16.12	1.99	1.16	814.75 ± 42.07	5.16	1.84	812.76 ± 13.51	1.66	1.59	812.27 ± 25.94	3.19	1.53
LIG	10	8.63 ± 1.08	12.49	−13.72	9.07 ± 0.82	9.01	−9.26	10.55 ± 0.42	4.01	5.47	9.42 ± 0.09	0.92	−5.84
	100	115.81 ± 2.42	2.09	15.81	113.35 ± 5.88	5.19	13.35	112.69 ± 2.42	2.15	12.69	113.95 ± 5.18	4.54	13.95
	800	830.05 ± 32.01	3.86	3.76	840.54 ± 47.91	5.70	5.07	859.33 ± 29.24	3.40	7.42	843.31 ± 36.92	4.38	5.41
BDPH	10	8.42 ± 0.09	1.03	−15.84	8.59 ± 0.47	5.45	−14.11	9.80 ± 0.90	9.15	−2.00	8.93 ± 0.10	1.08	−10.65
	100	93.06 ± 6.86	7.37	−6.94	94.52 ± 6.71	7.10	−5.48	92.26 ± 5.85	6.34	−7.74	93.28 ± 6.19	6.63	−6.72
	800	691.98 ± 36.60	5.29	−13.50	722.03 ± 17.34	2.40	−9.75	714.55 ± 27.37	3.83	−10.68	709.52 ± 29.66	4.18	−11.31
CYT	10	8.54 ± 0.31	3.62	−14.56	8.99 ± 0.48	5.31	−10.10	9.87 ± 0.84	8.49	−1.35	9.13 ± 0.10	1.05	−8.67
	100	100.69 ± 4.60	4.57	0.69	99.43 ± 2.37	2.38	−0.57	101.27 ± 3.73	3.69	1.27	100.46 ± 3.55	3.53	0.46
	800	840.36 ± 14.60	1.74	5.05	836.68 ± 18.23	2.18	4.59	842.71 ± 23.42	2.78	5.34	839.92 ± 18.69	2.22	4.99
NKT	10	8.60 ± 0.39	4.56	−13.97	8.46 ± 0.48	5.69	−15.44	8.85 ± 0.08	0.95	−11.51	8.60 ± 0.08	0.89	−13.98
	100	98.23 ± 3.40	3.46	−1.76	98.99 ± 3.66	3.69	−1.01	101.72 ± 3.81	3.75	1.72	99.65 ± 3.68	3.69	−0.35
	800	831.78 ± 19.20	2.31	3.97	809.02 ± 32.33	3.98	1.13	819.4 ± 8.84	1.08	2.42	820.07 ± 22.64	2.76	2.51
CYP	10	8.49 ± 0.55	6.45	−15.11	8.67 ± 0.49	5.63	−13.29	9.27 ± 0.86	9.26	−7.29	8.81 ± 0.05	0.58	−11.90
	100	97.75 ± 1.32	1.35	−2.25	101.08 ± 1.27	1.26	1.08	99.85 ± 1.74	1.74	−0.15	99.56 ± 1.97	1.98	−0.44
	800	816.27 ± 24.25	2.97	2.03	839.54 ± 34.55	4.12	4.94	835.68 ± 43.43	5.20	4.46	830.5 ± 34.47	4.15	3.81

**Table 2 molecules-24-02230-t002:** Intra-and inter-day precisions and accuracies of the analytes in rat cerebral cortex homogenate (*n* = 6).

		Day 1			Day 2			Day 3			Inter-day		
Analytes	Spiked Conc. (ng/mL)	Mean±SD (ng/mL)	RSD(%)	RE(%)	Mean±SD (ng/mL)	RSD(%)	RE(%)	Mean±SD (ng/mL)	RSD(%)	RE(%)	Mean±SD (ng/mL)	RSD(%)	RE(%)
SA	10	10.20 ± 0.33	3.25	2.02	10.20 ± 0.36	3.59	1.97	9.27 ± 0.21	2.25	−7.34	9.89 ± 0.30	3.02	−1.12
	100	107.11 ± 4.87	4.55	7.11	94.00 ± 2.24	2.38	−6.00	92.26 ± 3.27	3.55	−7.74	97.79 ± 3.46	3.49	−2.21
	800	770.88 ± 27.38	3.55	−3.64	767.86 ± 32.04	4.17	−4.02	703.46 ± 25.53	3.63	−12.07	747.40 ± 28.32	3.78	−6.57
NBP	10	9.81 ± 0.35	3.59	−1.90	10.64 ± 0.45	4.25	6.42	9.42 ± 0.22	2.32	−5.76	9.96 ± 0.34	3.39	−0.41
	100	106.47 ± 3.18	2.99	6.47	94.27 ± 2.12	2.24	−5.73	92.39 ± 2.62	2.83	−7.61	97.71 ± 2.64	2.69	−2.29
	800	768.72 ± 33.47	4.35	−3.91	771.41 ± 35.18	4.56	−3.57	714.29 ± 36.62	5.13	−10.71	751.47 ± 35.09	4.68	−6.07
LIG	10	9.38 ± 0.37	3.95	−6.15	9.73 ± 0.28	2.91	−2.74	8.73 ± 0.32	3.66	−12.7	9.28 ± 0.32	3.51	−7.20
	100	106.81 ± 4.83	4.52	6.81	93.92 ± 2.24	2.38	−6.08	90.81 ± 3.13	3.45	−9.19	97.18 ± 3.40	3.45	−2.82
	800	809.52 ± 32.48	4.01	1.19	803.00 ± 42.98	5.35	0.37	731.19 ± 31.90	4.36	−8.6	781.24 ± 35.78	4.58	−2.35
BDPH	10	9.70 ± 0.56	5.74	−2.98	10.26 ± 0.28	2.77	2.61	8.78 ± 0.37	4.25	−12.24	9.58 ± 0.40	4.25	−4.21
	100	102.51 ± 3.12	3.05	2.51	93.16 ± 2.79	3.00	−6.84	89.32 ± 3.26	3.65	−10.68	95.00 ± 3.06	3.23	−5.00
	800	770.61 ± 25.70	3.33	−3.67	793.52 ± 35.03	4.41	−0.81	719.14 ± 24.01	3.34	−10.11	761.09 ± 28.24	3.70	−4.86
CYT	10	10.00 ± 0.28	2.81	−0.02	10.40 ± 0.39	3.75	4.02	9.85 ± 0.23	2.31	−1.52	10.08 ± 0.30	2.96	0.83
	100	106.11 ± 4.24	4.00	6.11	94.56 ± 2.94	3.11	−5.44	96.76 ± 3.46	3.58	−3.24	99.15 ± 3.55	3.56	−0.85
	800	781.57 ± 25.55	3.27	−2.30	776.37 ± 44.71	5.76	−2.95	762.13 ± 39.57	5.19	−4.73	773.36 ± 36.61	4.74	−3.33
CYP	10	8.56 ± 0.94	10.98	−14.42	8.72 ± 0.74	8.45	−12.80	9.25 ± 1.09	11.75	−7.49	8.84 ± 0.92	10.39	−11.57
	100	96.88 ± 3.86	3.98	−3.12	87.57 ± 2.30	2.62	−12.43	84.99 ± 4.46	5.25	−15.00	89.82 ± 3.54	3.95	−10.18
	800	751.52 ± 21.48	2.86	−6.06	798.13 ± 50.76	6.36	−0.23	726.66 ± 40.08	5.52	−9.17	758.77 ± 37.44	4.91	−5.15

**Table 3 molecules-24-02230-t003:** Matrix effects and recoveries of the analytes in rat serum (*n* = 3).

		Matrix Effect		Recovery	
Analytes	Spiked Conc. (ng/mL)	Mean ± SD (ng/mL)	RSD(%)	Mean ± SD (ng/mL)	RSD(%)
FA	10	116.61 ± 2.78	2.38	85.29 ± 2.11	2.47
	100	104.11 ± 5.46	5.24	95.27 ± 5.80	6.09
	800	98.58 ± 3.40	3.45	102.51 ± 3.92	3.82
SA	10	104.60 ± 3.42	3.27	97.34 ± 2.38	2.44
	100	101.09 ± 4.93	4.88	104.04 ± 0.67	0.64
	800	101.34 ± 1.39	1.37	100.42 ± 2.57	2.56
NBP	10	99.28 ± 1.08	1.09	100.50 ± 1.24	1.24
	100	93.90 ± 0.71	0.76	105.39 ± 2.08	1.98
	800	99.78 ± 1.46	1.46	99.41 ± 4.44	4.47
LIG	10	99.71 ± 0.22	0.22	99.74 ± 0.07	0.07
	100	97.98 ± 2.85	2.91	102.63 ± 1.89	1.85
	800	97.17 ± 1.27	1.30	102.78 ± 2.42	2.36
BDPH	10	98.33 ± 0.51	0.52	102.54 ± 0.94	0.91
	100	94.16 ± 2.24	2.38	103.14 ± 5.51	5.34
	800	95.99 ± 3.99	4.16	95.94 ± 4.32	4.50
CYT	10	95.23 ± 0.94	0.99	108.49 ± 2.88	2.65
	100	97.05 ± 0.48	0.49	102.68 ± 1.48	1.44
	800	97.35 ± 4.56	4.68	103.61 ± 1.59	1.53
NKT	10	97.85 ± 2.41	2.46	101.40 ± 0.47	0.46
	100	101.34 ± 2.87	2.84	102.43 ± 3.18	3.10
	800	101.44 ± 2.61	2.57	101.65 ± 2.85	2.80
CYP	10	104.94 ± 2.09	1.99	97.19 ± 2.22	2.28
	100	97.46 ± 3.43	3.52	103.91 ± 2.26	2.17
	800	104.63 ± 1.53	1.46	98.44 ± 2.74	2.78

**Table 4 molecules-24-02230-t004:** Matrix effects and recoveries of the analytes in rat cerebral cortex homogenate (*n* = 3).

		Matrix Effect		Recovery	
Analytes	Spiked Conc. (ng/mL)	Mean ± SD (ng/mL)	RSD(%)	Mean ± SD (ng/mL)	RSD(%)
SA	10	92.55 ± 0.32	0.35	86.75 ± 0.80	0.92
	100	94.89 ± 2.90	3.06	97.16 ± 6.39	6.58
	800	102.12 ± 3.35	3.28	98.70 ± 1.43	1.45
NBP	10	96.96 ± 0.92	0.95	86.95 ± 10.17	11.69
	100	94.83 ± 1.82	1.92	88.12 ± 13.12	14.89
	800	103.36 ± 1.42	1.38	98.21 ± 2.20	2.24
LIG	10	99.54 ± 3.49	3.51	87.34 ± 2.79	3.19
	100	95.32 ± 2.38	2.50	101.33 ± 3.86	3.81
	800	100.10 ± 1.91	1.91	98.33 ± 3.11	3.17
BDPH	10	102.78 ± 6.59	6.41	87.74 ± 12.13	13.82
	100	97.30 ± 1.85	1.90	89.55 ± 12.28	13.72
	800	101.85 ± 2.15	2.11	98.00 ± 2.16	2.21
CYT	10	87.30 ± 3.77	4.32	97.28 ± 12.93	13.29
	100	95.68 ± 1.19	1.25	91.29 ± 10.96	12.01
	800	99.56 ± 3.53	3.55	100.20 ± 2.06	2.06
CYP	10	97.27 ± 2.95	3.03	84.96 ± 9.73	11.46
	100	86.62 ± 11.35	13.11	102.39 ± 9.33	9.12
	800	101.45 ± 2.03	2.00	107.23 ± 13.05	12.17

**Table 5 molecules-24-02230-t005:** Stability results of the analytes in rat serum under three different conditions (*n* = 3).

		Room Temperature for 24 h			At −80 °C for 30 Days			Three Freeze-thaw Cycles		
Analytes	Spiked Conc. (ng/mL)	Mean ± SD (ng/mL)	RSD (%)	RE (%)	Mean ± SD (ng/mL)	RSD (%)	RE (%)	Mean ± SD (ng/mL)	RSD (%)	RE (%)
FA	10	8.79 ± 0.27	3.04	−12.11	8.77 ± 0.35	3.97	−12.34	8.40 ± 0.15	1.83	−15.99
	100	85.72 ± 5.69	6.64	−14.28	87.07 ± 4.59	5.27	−12.93	89.87 ± 2.25	2.50	−10.13
	800	759.35 ± 29.99	3.95	−5.08	802.77 ± 11.52	1.43	0.35	729.41 ± 12.38	1.70	−8.82
SA	10	11.22 ± 0.43	3.82	12.24	10.54 ± 0.22	2.07	5.38	11.16 ± 0.61	5.47	11.62
	100	106.01 ± 1.91	1.80	6.01	108.86 ± 0.82	0.75	8.86	105.69 ± 3.65	3.45	5.69
	800	806.11 ± 27.70	3.44	0.76	801.30 ± 17.63	2.20	0.16	816.77 ± 7.32	0.90	2.10
NBP	10	9.77 ± 0.06	0.57	−2.35	9.87 ± 0.09	0.87	−1.32	9.71 ± 0.06	0.58	−2.87
	100	111.97 ± 1.98	1.76	11.97	113.29 ± 1.60	1.41	13.26	107.04 ± 2.83	2.64	7.04
	800	838.88 ± 11.37	1.36	4.86	830.25 ± 21.08	2.53	4.16	830.01 ± 21.06	2.54	3.75
LIG	10	10.68 ± 0.75	7.02	6.76	10.41 ± 0.62	5.97	4.10	9.20 ± 0.81	8.83	−7.95
	100	111.69 ± 3.33	2.98	11.69	114.18 ± 1.70	1.49	14.18	110.57 ± 2.73	2.47	10.57
	800	804.52 ± 10.70	1.33	0.56	813.62 ± 8.03	0.99	1.70	811.95 ± 11.88	1.46	1.49
BDPH	10	11.05 ± 0.60	5.42	10.50	11.34 ± 0.26	2.29	13.44	10.70 ± 0.17	1.56	7.00
	100	100.05 ± 2.12	2.11	0.05	104.30 ± 0.98	0.94	4.30	103.1 ± 4.50	4.36	3.10
	800	754.77 ± 28.27	3.75	−5.65	759.20 ± 9.40	1.24	−5.10	770.56 ± 8.90	1.16	−3.68
CYT	10	9.78 ± 0.08	0.83	−2.17	9.86 ± 0.35	3.59	−1.42	9.79 ± 0.09	0.88	−2.05
	100	103.59 ± 3.83	3.69	3.59	99.21 ± 3.23	3.25	−0.79	101.18 ± 4.77	4.71	1.18
	800	853.39 ± 33.91	3.97	6.67	866.15 ± 16.88	1.95	8.27	854.54 ± 6.58	0.77	6.82
NKT	10	9.83 ± 0.11	1.08	−1.66	9.67 ± 0.04	0.40	−3.32	9.66 ± 0.07	0.75	−3.38
	100	104.65 ± 4.89	4.68	4.65	103.29 ± 1.58	1.53	3.29	101.77 ± 2.21	2.17	1.77
	800	799.56 ± 16.00	2.00	−0.06	830.96 ± 2.56	0.31	3.87	808.31 ± 31.1	3.85	1.04
CYP	10	10.37 ± 0.12	1.20	3.69	9.93 ± 0.03	0.30	−0.73	10.41 ± 0.09	0.85	4.06
	100	107.47 ± 4.16	3.87	7.47	105.42 ± 1.41	1.34	5.42	106.74 ± 1.81	1.70	6.74
	800	819.35 ± 18.63	2.27	2.42	854.21 ± 25.57	2.99	6.78	858.57 ± 19.45	2.27	7.32

**Table 6 molecules-24-02230-t006:** Stability results of the analytes in rat cerebral cortex homogenate under three different conditions (*n* = 3).

		Room Temperature for 24 h			At −80 °C for 30 Days			Three Freeze-thaw Cycles		
Analytes	Spiked Conc. (ng/mL)	Mean ± SD (ng/mL)	RSD(%)	RE(%)	Mean ± SD (ng/mL)	RSD(%)	RE(%)	Mean ± SD (ng/mL)	RSD(%)	RE(%)
SA	10	11.57 ± 0.34	2.94	15.71	11.78 ± 0.36	3.05	17.76	11.36 ± 0.49	4.31	13.62
	100	94.24 ± 3.37	3.58	−5.76	118.32 ± 3.48	2.94	18.32	106.00 ± 13.18	12.43	6.00
	800	695.93 ± 16.75	2.41	−13.01	784.94 ± 58.92	7.51	−1.88	699.79 ± 22.43	3.20	−12.53
NBP	10	11.79 ± 0.83	7.06	17.94	11.76 ± 0.49	4.18	17.60	11.34 ± 0.63	5.58	13.41
	100	96.12 ± 4.77	4.96	−3.88	116.79 ± 3.38	2.89	16.79	106.56 ± 12.70	11.92	6.56
	800	690.98 ± 12.03	1.74	−13.63	791.92 ± 71.99	9.09	−1.01	707.28 ± 27.04	3.82	−11.59
LIG	10	11.55 ± 0.39	3.41	15.45	11.83 ± 0.54	4.58	18.31	11.66 ± 0.44	3.74	16.55
	100	94.22 ± 3.24	3.44	−5.78	117.34 ± 2.70	2.30	17.34	99.84 ± 11.11	11.13	−0.16
	800	754.92 ± 2.98	0.39	−5.63	800.66 ± 75.63	9.45	0.08	744.87 ± 35.58	4.78	−6.89
BDPH	10	11.34 ± 0.82	7.24	13.37	9.36 ± 1.43	15.28	−6.42	8.45 ± 1.46	17.31	−15.47
	100	86.68 ± 1.94	2.24	−13.32	108.47 ± 4.73	4.36	8.47	97.54 ± 10.22	10.47	−2.46
	800	754.64 ± 10.13	1.34	−5.67	760.72 ± 19.30	2.54	−4.91	751.83 ± 36.02	4.79	−6.02
CYT	10	11.32 ± 0.59	5.19	13.21	11.66 ± 0.40	3.41	16.60	11.34 ± 0.14	1.23	13.44
	100	91.59 ± 2.13	2.33	−8.41	115.20 ± 1.63	1.41	15.20	104.71 ± 11.23	10.73	4.71
	800	720.43 ± 17.57	2.44	−9.95	833.54 ± 64.02	7.68	4.19	717.10 ± 36.67	5.11	−10.36
CYP	10	10.98 ± 0.51	4.64	9.84	11.88 ± 0.30	2.55	18.76	11.17 ± 0.55	4.96	11.74
	100	86.69 ± 3.07	3.54	−13.31	114.96 ± 4.89	4.25	14.96	105.74 ± 13.31	12.59	5.74
	800	665.61 ± 38.39	5.77	−16.80	783.49 ± 58.64	7.49	−2.06	678.78 ± 13.11	1.93	−15.15

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
