# Peer review of "Anti-Migraine Effect of the Herbal Combination of Chuanxiong Rhizoma and Cyperi Rhizoma and UPLC-MS/MS Method for the Simultaneous Quantification of the Active Constituents in Rat Serum and Cerebral Cortex"

_molecules, 2019, doi:10.3390/molecules24122230_

Round 1

Reviewer 1 Report

 This work is very nice and interesting. I have only doubts regarding the quality of plant material. The authors should provide more information about their origin, quality characteristics, standardisation of extracts.

Also abstract is too long and confusing. I would suggest to leave only the most important information in it.

Author Response

The responses were showed in the word file.

Reviewer 2 Report

The work taking into consideration two traditional Chinese herbs (Chuanxiong Rhizoma and Cyperi Rhizoma) used for curing migraine. The subject is important and interesting because of the fact that more and more people suffer from migraine. That is why it is important to constantly search for new drugs against this ailment. The special attention is paid to the combination of Chuanxiong Rhizoma and Cyperi Rhizoma in the presented work. Many active compounds were analyzed by use of UPLC-MS/MS and there was a discussion carried out about their content after administration during a migraine attack. The introduction of the work is exhaustive, supported by appropriate literature. The UPLC-MS/MS method was optimized and validated. All results are clearly presented in figures and tables.  The method of analysis is also clearly presented and could be easy duplicate. A discussion supported by literature describes in detail the influence of active substances on the course of a migraine attack. The study confirmed that the combination of two herbs gives the preventive and therapeutic effects of migraine. Other clinical trials would be worthwhile because, as it is commonly known, migraine has a different etiology and only artificially induced nitroglycerin is investigated in the presented work.

Author Response

(The authors gave the same response as above.)

Reviewer 3 Report

This is a data-rich manuscript. The reviewer has some comments:

(1) In Fig 1, why CRCR decreased cerebral blood flow in comparison to control?

(2) For LC-MS/MS analysis, the assays for the phyto-chemicals may have been reported before. Please quote the references properly.

(3) In abstract, how can the authors conclude:  "The active constituents in CRCR for treating migraine were FA, SA, NBP, LIG, BDPH, CYT, NKT and CYP. SA, NBP, LIG, BDPH, CYT ..."?

Author Response

(The authors gave the same response as above.)

Reviewer 4 Report

The effect of an herbal extract currently used in traditional chinese medicine for migraine soothing is studied in regard to its chemical components. Chuanxiong Rhizoma and Cyperi Rhizoma (CRCR) are the main herbal components of the formule. Diverse biochemical parameters related to the migraine are also analyzed as indicators of the pathology and the herbal extract effect.

The study is well conducted and reported. A couple of issues should be addressed before publication:

1) CRCR is used in the study at a proportion of 1:2 for Chuanxiong Rhizoma and Cyperi Rhizoma; however, the authors point out that the formule for migraine treatment currently account for other herbs, and consequently a dilution effect can be expected. Could the authors explain how the dosage used in this study accommodates to the popular formulation?

2) Could the authors explain a bit better which represents the values on the column "Mean+-SD" of "Matrix effects" in tables 3 and 4?    

Author Response

(The authors gave the same response as above.)

Round 2

Reviewer 1 Report

I have no more comments, authors corrected manuscript according to provided suggestions.